# The Effects of Nitrogen Fertilizer on the Aroma of Fresh Tea Leaves from *Camellia sinensis cv.* Jin Xuan in Summer and Autumn

**DOI:** 10.3390/foods13111776

**Published:** 2024-06-05

**Authors:** Ansheng Li, Zihao Qiu, Jinmei Liao, Jiahao Chen, Wei Huang, Jiyuan Yao, Xinyuan Lin, Yuwang Huang, Binmei Sun, Shaoqun Liu, Peng Zheng

**Affiliations:** 1College of Horticulture, South China Agricultural University, Guangzhou 510642, China; las1533854642@stu.scau.edu.cn (A.L.); scau20222018004@stu.scau.edu.cn (Z.Q.); cjhtea@stu.scau.edu.cn (J.C.); 13502975421huangwei_chris@163.com (W.H.); yaojuan@stu.scau.edu.cn (J.Y.); imxyuanlin@stu.scau.edu.cn (X.L.); hlbs@stu.scau.edu.cn (Y.H.); binmei@scau.edu.cn (B.S.); scauok@scau.edu.cn (S.L.); 2Soiland Fertilizer Station of Cenxi City, Wuzhou 543200, China; ljm19127614165@stu.scau.edu.cn

**Keywords:** *Camellia sinensis*, nitrogen fertilizer, harvesting season, fresh tea leaves, volatile compounds

## Abstract

Nitrogen fertilization level and harvesting season significantly impact tea aroma quality. In this study, we analyzed the volatile organic compounds of fresh Jin Xuan (JX) tea leaves under different nitrogen application levels (N0, N150, N300, N450) during summer and autumn. A total of 49 volatile components were identified by gas chromatography–mass spectrometry (GC-MS). Notably, (E)-2-hexenal, linalool, and geraniol were the main contributors to the aroma of fresh JX leaves. The no-nitrogen treatment (N0) presented the greatest quantity and variety of volatiles in both seasons. A greater difference in volatile compounds was observed between nitrogen treatments in summer vs. autumn. The N0 treatment had a greater total volatile concentration in summer, while the opposite was observed in the nitrogen application treatments (N150, N300, N450). Summer treatments appeared best suited to black tea production. The concentration of herbaceous aroma-type volatiles was higher in summer, while the concentration of floral volatiles was higher in autumn. Volatile concentrations were highest in the N0 and N450 treatments in autumn and appeared suitable for making black tea and oolong tea. Overall, this research provides valuable insights into how variations in N application rates across different harvesting seasons impact the aroma characteristics of tea leaves.

## 1. Introduction

Tea (*Camellia sinensis* L.) is a woody perennial evergreen plant primarily cultivated in humid regions of the tropics, subtropics, and temperate zones. Young buds and leaves are harvested for their health benefits and pleasant flavor, making it one of the most popular non-alcoholic caffeinated beverages [1,2]. The Jinxuan cultivar (JX) has the characteristics of high yield, fertilizer tolerance, and excellent quality. The JX produces fresh leaves that are used to make high-quality green and oolong tea in spring, and oolong and black tea in summer and autumn [3].

Aroma serves as a pivotal indicator of tea quality, playing a crucial role in its sensory attributes [1,4]. Over 700 volatile compounds have been identified in tea and its brews to date [5,6]. The enzymatic and chemical properties of harvested tea leaves are affected by seasonal climactic changes such as temperature, humidity, precipitation, and solar irradiance [7,8]. Changes in enzymatic conversion and degradation of metabolic precursors are primarily responsible for variations in aroma composition and content observed among teas harvested in different seasons [9,10,11,12]. Thus, harvesting season significantly influences tea aroma quality.

In addition to seasonality, plant fertilization may also play a pivotal role in tea aroma quality. Extensive fertilization is recommended from summer to autumn. Nitrogen (N) plays a pivotal role in tea plant growth and development as a key component of proteins, nucleic acids, phospholipids, chlorophyll, and hormones essential for tea plant growth and development and is closely tied to tea quality [13]. Studies suggest that appropriate N fertilizer application regulates lipid metabolism, enhances aroma compound formation, and improves tea leaf quality [14]. However, excessive N fertilization may elevate herbaceous flavor precursors in tea, potentially diminishing tea aroma quality [15]. Notably, most aroma compounds are secondary products of carbon (C) metabolism, with C and N metabolism in plants intricately linked and competitive, aligning with the “carbon-nutrient balance” hypothesis proposed by Bryant et al. [16].

Lipids in tea leaves are believed to contribute significantly to flavor compound production, while oxidation products of free fatty acids formed during tea processing, such as (Z)-3-hexenol and (E)-2-hexenal, act as aroma compound precursors, enhancing tea freshness and aroma [17]. Furthermore, characteristic volatiles of fresh leaves also influence tea leaf processing suitability [18].

In this study, we employed gas chromatography coupled with headspace solid-phase microextraction to examine the aroma profile of JX tea leaves. By utilizing partial least squares regression, we delved into the significance of projected variables, iso-multivariate, and odor activity values to elucidate the fluctuating composition of volatiles in JX tea leaves subjected to various N treatments and harvested during different seasons. Furthermore, we investigated the correlations among these variables and their implications for subsequent processing (Figure 1). Presently, the interaction effects of harvest season and N fertilization level on the dynamic changes of volatiles and processing suitability of JX tea remain unclear, necessitating a systematic investigation into the comparative effects of different N fertilization levels applied during summer and autumn. It is expected that these data will contribute scientific insights into how variations in N application rates across different harvesting seasons impact the aroma characteristics of tea leaves and processing suitability (Figure 1).

## 2. Materials and Methods

### 2.1. Chemicals and Reagents

Alkane standard solutions (C8-C40) for calculating linear retention indices (RIs) were supplied by the Beijing Tianmo Quality Testing Technology Co. (Beijing, China). Prior to usage, the internal standard solution was prepared using methylene chloride. Ultrapure water (type 1) was produced using a Barnstead GenPure Pro system (Thermo Fisher Scientific, Waltham, MA, USA).

### 2.2. Experimental Design and Collection of Tea Samples

The experimental tea plantation was situated at South China Agricultural University (23.16° N, 113.36° E), Guangzhou City, Guangdong Province, China. The JX tea plant variety was used, with urea fertilizer chosen for N application. The experiment was arranged into four treatments, each following a randomized complete block design (RCBD) with three replicates. Each plot measured 1.2 square meters (1.2 m × 1 m). Treatments received a total N application of 0 kg/hm^2^ (N0), 150 kg/hm^2^ (N150), 300 kg/hm^2^ (N300), and 450 kg/hm^2^ (N450). The specific timing and proportion of fertilizer application were as follows: 30% at the end of April 2022 (i.e., basal fertilizer), 20% by the end of June 2022 (i.e., summer fertilizer I), 20% by the end of August 2022 (i.e., summer fertilizer II), and 30% by the end of October 2022 (i.e., autumn fertilizer). Fertilizers were applied in strip trenches approximately 20–30 cm from the base of tea plants at a depth of 15–20 cm, and were then covered with soil. Fresh tea samples, comprising one bud and two leaves, were gathered in early June and early October 2022 from all tea plants. These samples were promptly frozen in liquid nitrogen and preserved in a −80 °C ultra-low-temperature freezer. Leaf collection was randomized, with three biological replicates per sample group.

### 2.3. Extraction of Volatiles Utilizing Head Space Solid-Phase Microextraction Gas Chromatography (HS-SPME GC)

HS-SPME GC was employed for the identification and quantification of volatiles. An Agilent 1890B gas chromatograph coupled with a 5977A mass spectrometer (Agilent, Santa Clara, CA, USA) equipped with an HP-5 MS column (30 m × 0.25 mm × 0.25 μm film thickness) was utilized. An SPME device comprising divinylbenzene/carboxenic acid/polydimethylsiloxane (DVB/CAR/PDMS) fibers (50/30 μm inner diameter, 2 cm length; Supelco, Darmstadt, Germany) was introduced into the headspace vials. Approximately 0.2 g of each sample was homogenized and accurately weighed into 50 mL vials. The 50/30 μm DVB/Carboxen/PDMS fiber (Supelco, St. Louis, MO, USA) was inserted into the headspace of the vial and maintained for 40 min to ensure the desorption of volatiles by the SPME fiber. Compounds were subsequently extracted at 80 °C for 40 min. Following extraction, the SPME fiber was inserted into the GC-MS at 250 °C for 3 min.

Each measurement was taken three times, and the values were averaged.

### 2.4. GC-MS Analysis of Volatile Compounds

The column flow rate was set at 1.0 mL/min with high-purity helium (purity ≥ 99.99%) serving as the carrier gas, with a solvent delay time of 4 min. The initial temperature of the ion source was held at 50 °C for 1 min, ramped to 220 °C at 5 °C/min, and held for 5 min. The temperature of the ion source was maintained at 230 °C, and the electron impact (EI) ionization source operated at 70 eV. The scan range spanned 30 to 400 amu.

Quantification was conducted using established methodologies. Briefly, the peak area ratio of the internal standard to the target compound was calculated to determine the actual concentration of the target compound based on its concentration ratio. Compound identification was accomplished by cross-referencing the National Institute of Standards and Technology (NIST, https://webbook.nist.gov/ (accessed on 13 December 2023)) mass spectrometry database with compounds’ retention indices (RIs), which were determined using n-alkanes C9–C21. The RI was calculated as follows:RI = 100n + 100 [RT(x) − RT(n)]/[RT (n + 1) − RT(n)]
where RT(x) represents the retention time of compound x, and RT(n) and RT(n + 1) denote the retention times of alkanes with carbon numbers n and n + 1 that immediately preceded and followed the elution of the compound.

### 2.5. Calculation of Odor Activity Values (OAVs)

The OAV represents the ratio of the concentration of an aroma component to its odor threshold in an aqueous solution. OAVs were employed to assess the contributions of each volatile compound to the perceptible aroma of the tea samples. The OAV of each volatile compound (OAVi) was calculated as OAVi = Ci/Ti, where Ci represents the concentration of compound i (μg kg^−1^) and Ti is the threshold value of compound i (μg kg^−1^).

### 2.6. Statistical Analysis

Raw data from three replicate experiments were analyzed using Microsoft Excel 2021. One-way analysis of variance (ANOVA) was performed utilizing the SPSS 27 software package (SPSS Inc., Chicago, IL, USA) to assess significant differences between samples. GraphPad Prism 9.5.0 (GraphPad Software, Inc., La Jolla, CA, USA) was employed for additional analyses. SIMCA (Version 14.1, Umetrics, Umea, Sweden) software constructed plots for orthogonal partial least squares discriminant analyses (PLS-DA) and variable importance in projection (VIP). A variable with a VIP score ≥ 1 predicts that a variable is important in the model.

## 3. Results and Discussion

### 3.1. Identification and Comprehensive Comparison of Volatiles

In this study, we analyzed the volatiles present in fresh JX tea leaves grown under four N application levels (N0, N150, N300, and N450) during summer and autumn. A total of forty-nine volatile compounds were identified and categorized into eight groups based on their functional groups: ten alcohols, five aldehydes, ten ketones, eleven alkenes, four esters, three hydrocarbons, two heterocyclic compounds, and four other compounds (Figure 2A). To study their relative contributions in teas grown in the different treatments, the volatile components were quantified (Appendix A). Base peak chromatogram (BPC) analysis confirmed that all samples demonstrated outstanding capability for mass spectrometry signal detection, boasting a wide peak capacity and significant separation efficiency (Appendix A).

Figure 2B illustrates that the predominant aroma profiles of fresh JX leaves are grass and floral notes, complemented by fruity, sweet, and woody undertones. Aroma compounds were predominantly aldehydes and alcohols across all treatments (Figure 2C–E). Additionally, 11 and 17 volatile compounds were shared across different N application treatments during summer and autumn, respectively. These findings align with previous studies [19,20].

Aldehydes represent a significant source of tea aroma, comprising the largest proportion of total volatiles in most green, oolong, and black teas [21]. While most aldehydes in tea impart citrus and green flavors, some contribute bitter almond, malt, honey, fatty, nutty, and even metallic notes [5]. Aldehydes in tea are formed through two main pathways: lipid oxidation mediated by lipid oxidases and Strecker degradation [21]. (E)-2-hexenal serves as the primary volatile substance in oolong tea, originating from the oxidation of free fatty acids during the tea manufacturing process, contributing to the tea leaves’ freshness and aroma [22]. Decanal, another aldehyde, typically derives from the oxidation products of fatty acids and emits a robust oily, citrus, and floral scent.

Alcohols are also prominent volatiles in tea aroma [23], constituting approximately 50% of total volatiles in most black teas [24]. Linalool and geraniol, terpene alcohols, are derived from geranyl pyrophosphate precursors through geraniol synthase and linalool synthase, respectively. Linalool, prevalent in various tea varieties, imparts a characteristic sweet floral aroma, playing a crucial role as a volatile compound [25]. Geraniol, with its aromatic and sweet flavor, emits floral and citrusy aromas, with subtle herbal and sweet undertones. The variations in concentration may be attributed to nitrogen application treatments affecting geraniol or geranyl pyrophosphate, a precursor of linalool [5]. Methyl salicylate, an ester, is typically generated through the reaction of salicylic acid with methanol, possessing a minty fragrance [26].

We used cluster analysis to draw heat maps of the top ten high-concentration volatile compounds in fresh JX leaves (Appendix A). Among the top 10 volatile compounds with the highest relative concentrations were linalool, geraniol (E)-2-hexenal, decanal, 2,6-bis(1,1-dimethylethyl)-4-hydroxy-4-methyl-2,5-cyclohexadien-1-one, 2,6,10,14-tetramethylhexadecane, methyl salicylate, butylated hydroxytoluene; 4-(2,6,6-trimethyl-1-cyclohexen-1-yl)-3-buten-2-one, and β-myrcene, most of which have been identified in previous studies as key volatile constituents in tea leaves [27].

Our results indicate that the quantity and composition of volatiles in JX leaves are influenced by both the level of N fertilizer application and harvest season. Studies have demonstrated that processing fresh JX leaves into black tea [28] and oolong tea [29] results in changes in volatile composition, with a decrease in aldehyde content and an increase in alcohol content. These changes are primarily attributed to alterations in volatile enzymes and aroma precursors during processing [9].

### 3.2. Comparison of the Volatiles Found in Fresh JX Leaves in Summer and Autumn under the Same N Treatment

The total concentration of volatiles was significantly higher in summer than in autumn in the N0 treatment (Figure 2C). Conversely, the total volatile concentration in the N150, N300, and N450 treatments was lower in summer than in autumn. The results of the N-supplemented treatments reflect previous studies that observed higher volatile contents of black tea produced in autumn compared to that grown in spring and summer [30]. Notably, there was a significant difference in the total concentration of volatiles between the N0 treatment and the N treatments in summer, whereas no significant difference was observed in autumn.

As illustrated in Figure 2D, the primary categories of volatiles in both summer and autumn were aldehydes and alcohols, with aldehydes exhibiting higher concentrations in autumn across all treatments. Conversely, alcohols displayed the opposite trend in most N treatments. Figure 2E,F reveal that 42 and 40 substance categories appeared in summer and autumn. The differences in category numbers among treatments were not statistically significant.

Figure 3 demonstrates that high-concentration volatiles, (E)-2-hexenal, linalool, and geraniol, exhibited higher concentrations in autumn than in summer in most N application treatments. Some substances displayed higher concentrations in summer than in autumn without N application treatment, consistent with the overall trend of substance content. This phenomenon was attributed to the favorable temperatures in different seasons for the development of aroma compounds and their precursors in fresh tea leaves [31].

Regarding aroma types, as shown in Figure 4, the overall concentration of floral and woody volatiles was higher in summer compared with autumn and was most pronounced in the N0 treatment. Conversely, the concentration of grassy aroma volatiles was higher in autumn, especially in the N150 treatment. The degree of overlap of aroma types in individual treatments, depicted in radar charts, was higher in autumn than in summer, while the differences were greater in summer. Therefore, tea volatile components are noticeably influenced by seasonality.

Zhang et al. [32] found that the types and amounts of aromatic substances in spring and autumn teas were lower than those in summer tea. After processing fresh JX leaves into black tea, the primary types of volatiles included alcohols, esters, and so forth, with the key aroma volatiles being geraniol, methyl salicylate, and linalool [31]. Similarly, for oolong tea, the main types of volatiles consist of alcohols, aldehydes, hydrocarbons, and others, with the predominant aroma volatiles being linalool, geraniol, and β-myrcene [27]. In this investigation, the characteristic volatiles of fresh leaves exhibited similarities with both black tea and oolong tea. Furthermore, the characteristic volatiles of black tea from fresh leaves harvested in summer surpassed those of autumn, indicating that JX harvested in summer is more suitable for black tea processing than that harvested in autumn, a finding consistent with prior research [30,33,34].

In summary, the application of N fertilizer in summer resulted in a reduction in volatiles, with the most significant decrease observed in the N300 treatment, followed by a rebound in the N450 treatment. In terms of volatile matter, summer JX is more suitable for processing into black tea.

### 3.3. Volatile Components of Fresh JX Leaves in Summer under the Different N Treatments

Under summer conditions, depicted in Figure 2C, the overall variation in total volatile content exhibited a decrease with increasing N fertilization from N0 to N300, followed by a rebound at N450. Aldehydes and alcohols constituted the main volatile substances found in all treatments, accounting for proportions ranging from 51.71% to 64.08% and 29.16% to 38.92%, respectively (Figure 2E). This trend mirrored that of the total volatile content (Figure 3), while the content of other substance types, like olefins, esters, and heterocyclic rings, also decreased with N application. A total of 42 volatiles were identified, with 30, 29, 28, and 24 substances detected in the N0, N150, N300, and N450 treatments, respectively. Moreover, 15 common substances, such as (E)-2-hexenal, linalool, and geraniol, were consistently present across treatments (Figure 2D).

Among the top ten highly concentrated volatiles, eight were observed under summer conditions. As illustrated in Figure 3 and Figure 4, the volatile concentration in the N0 treatment significantly exceeded that of the N-applied treatments. The trend observed in the N-applied treatments demonstrated a decline from the N0 to N300 treatment, followed by a rebound at the N450 treatment, mirroring the trend observed in the total content. Based on the predominant volatiles in black tea and oolong tea, and the variations in volatile compounds across N application treatments during summer, it appears that the N0 treatment was more conducive to the production of black tea and oolong tea.

Regarding the aroma profiles of fresh JX leaves grown during summer, as depicted in Figure 4, the N450 treatment exhibited the highest aroma concentration among the N treatments. Conversely, the N300 treatment showed the lowest aroma concentration, primarily characterized by a grassy aroma type, attributed to changes in substances like (E)-2-hexenal. Previous research suggests that N may influence the precursor substances of volatile compounds in tea leaves, consequently impacting the synthesis of aroma compounds [13].

In summary, the general trend of volatile compounds in fresh JX leaves during summer is a decrease from the N0 to N300 treatments, with an increase in the N450 treatment. Based on the types and concentrations of key volatile compounds, the N0 treatment appears to be more suitable for producing black tea and oolong tea during summer (Figure 5).

### 3.4. Volatile Components of Fresh JX Leaves in Autumn under the Different N Treatments

As depicted in Figure 2C, under autumn conditions, the total volatile content was higher in the N0 treatment than in the N treatments. However, there were no significant differences in content changes among the N150, N300, and N450 treatments. The N0 treatment exhibited the highest total volatiles concentration, attributed to significant diffusion of (E)-2-hexenal. Aldehydes and alcohols were the predominant substances across all treatments (Figure 2D), comprising 61.92% to 66.53% and 20.83% to 29.16%, respectively. The N0 treatment showed the highest concentration, while N150, N300, and N450 exhibited irregular trends (Figure 3). Comparing N0, N150, N300, and N450, hydrocarbon substances disappeared, and the percentage of ketones, alkenes, hydrocarbons, and others decreased to the lowest observed levels (Figure 2D).

Based on the main volatiles of black and oolong tea discussed previously and the volatile substance changes observed in each N application treatment during autumn, it can be concluded that the N0 and N450 treatments are more suitable for black and oolong tea production. A total of 40 volatile substances were detected in autumn, with 33, 25, 26, and 27 detected in the N0, N150, N300, and N450 treatments, respectively (Figure 2E). Among these, 17 common substances, including linalool and geraniol, were consistently present (Figure 2F). The highest number of volatile species was found under the N0 treatment, and most were alkenes and alcohols. Additionally, γ-muurolene, α-muurolene, τ-muurolol, (3r,6s)-2,2,6-trimethyl-6-vinyltetrahydro-2h-pyran-3-ol, and others were exclusively present in the N0 treatment. These compounds are derivatives of the muurolene class, characterized by woody, herbaceous, and aromatic odors. The absence of these aroma substances may be attributed to N fertilizer application increasing the plant’s C demand, affecting photosynthesis and impairing aroma compound synthesis. Excessive N fertilizer also reduces the plant’s antioxidant capacity, leading to oxidative damage to certain compounds [28,34].

As illustrated in Figure 4 and Figure 5, key compound trends in autumn were diverse, with the top five volatiles with high concentrations being (E)-2-hexenal, linalool, geraniol, 2,6-bis(1,1-dimethylethyl)-4-hydroxy-4-methyl-2,5-cyclohexadien-1-one, and 2,6,10,14-tetramethylhexadecane. Total volatile content varied similarly, with the highest levels observed in the N0 treatment and irregular trends across N application treatments. Under autumn conditions, as shown in Figure 4, the N0 treatment exhibited higher aroma concentrations than all N-applied treatments. Among the N-applied treatments, N150 showed a higher concentration of grass aroma, while N300 exhibited higher concentrations of floral, woody, and sweet aromas compared to other N-applied treatments (Figure 1). In this study, nitrogen impacts the aromatic compounds of fresh JX leaves. Furthermore, the quantity of nitrogen fertilizer used in tea production plays a crucial role in determining the quality and yield of tea leaves. Consequently, the next step involves evaluating the yield and quality of fresh JX leaves under varying nitrogen levels, offering insights for future rational fertilization practices [20].

### 3.5. Multivariate Analysis of Volatiles in Fresh JX Leaves in Summer

Based on the analysis of forty-two volatile compounds across four N application treatments during summer, differences and similarities among samples were assessed using an unsupervised principal component analysis (PCA). The PCA score plot (Figure 5A) illustrates a distinct separation of summer samples along PC2, which explains 73.2% of the total variance. While the N150, N300, and N450 treatments appeared relatively proximate on the plot, they were distinctly separated, indicating distinguishable individual characteristics. Conversely, the N0 treatment exhibited significant differences, showcasing distinct traits compared to the other treatments.

To discern volatile compounds across different N levels, partial least squares discriminant analysis (PLS-DA) was employed to determine variable importance in projection (VIP) values. The model’s reliability was confirmed through 200 permutation tests (summer: R2 = 0.0, 0.662, Q2 = −0.492, Figure 5B). Figure 5C shows that all four N application treatments are within the confidence interval and have good repeated clustering.

From the summer dataset, fifteen discriminating aroma compounds (VIP > 1) were identified, comprising five alcohols, four ketones, three alkenes, two aldehydes, and one ester (Table 1). Notably, (E)-2-hexenal (VOC_1), linalool (VOC_4), geraniol (VOC_11), (E)-2,6-dimethylocta-3,7-diene-2,6-diol (VOC_7), and 7,9-di-tert-butyl-1-oxaspiro(4,5)deca-6,9-diene-2,8-dione (VOC_41) ranked among the top five VIPs.

The contribution of these volatiles to the overall tea aroma depends not only on their concentration but also on their odor threshold. Consistent with the PCA model, greater similarity was observed among the N150, N300, and N450 treatments, while more significant differences were evident in the N0 treatment. These findings corroborated earlier analyses. Odor descriptions and odor activity values (OAVs) of the volatile compounds are presented in Table 1. It is widely acknowledged that compounds with OAV > 1 are aromatically active, with OAV values directly correlated to aroma contribution. Among the aroma OAV values listed in Table 1, nine species with VIP values exceeding 1 in fresh JX leaves were identified based on odor thresholds. Of these, eight volatiles exhibited an OAV > 1, with four contributing to floral attributes, two to fruity attributes, and two to herbaceous attributes. Notably, linalool and trans-β-ionone, with OAV > 1000, emerged as the primary contributors to the floral aroma of fresh JX leaves during summer. Specifically, trans-β-ionone imparts a violet floral aroma, present notably in the N0 and N150 treatments. β-ionone, a representative ketone among aroma-active compounds in tea, is reported at concentrations exceeding odor thresholds across various teas and is readily synthesized during tea withering [26,35].

**Table 1 foods-13-01776-t001:** Odor activity value (OAV) analysis of volatile compounds in fresh Jin Xuan leaves at four N application levels in summer.

Var. No.	Volatile Compounds	Odor Type	VIP ^1^	OT ^2^ (µg/kg)	OAVs in N Treatments			
					N0	N150	N300	N450
Var_1	(E)-2-hexenal	Fresh, fruity, green	3.22	110 ^a^	15.83	10.49	8.60	12.88
Var_4	Linalool	Floral, sweet	2.65	0.22 ^b^	4379.73	1842.73	1892.82	2460.45
Var_11	Geraniol	Rose-like, sweet, honey-like	2.11	0.6 ^a^	461.98	256.7	224.43	145.6
Var_9	Decanal	Citrus, fatty, green	1.37	9 ^a^	8.89	8.18	6.36	4.10
Var_8	Methyl salicylate	Peppermint, minty, fresh, sweet	1.27	40 ^a^	1.55	0.59	0.22	0.68
Var_32	(3R,6S)-2,2,6-trimethyl-6-vinyltetrahydro-2H-pyran-3-ol	Woody	1.25	3000 ^c^	0.00	0.01	0.01	0.01
Var_37	2,6,6-trimethyl-1-cyclohexene-1-carboxaldehyde	Herbal, clean, rose-like, fruity	1.07	5 ^d^	0.00	0.00	2.30	0.00
Var_12	α-Cubebene	Herbal waxy	1.02	14 ^e^	1.31	0.00	0.67	0.00
Var_20	trans-β-ionone	Violet	1.00	0.01 ^f,g^	2838.00	1380.00	0.00	0.00

^1^ VIP, variable importance in projection; ^2^ OT, odor threshold in water reported elsewhere, indicated as a–g [36,37,38,39,40,41,42].

### 3.6. Multivariate Analysis of Volatiles in Fresh JX Leaves in Autumn

Based on 40 volatile compounds, the samples underwent PCA to assess differences and similarities. The PCA score plot (Figure 5E) revealed a significant separation of samples in autumn along PC2, collectively explaining 56.2% of the total variance. Notably, N application treatments appeared closer to each other, with N0 exhibiting a distinct difference.

To predict the most important volatile compounds present in fresh JX leaves at varying N levels during autumn, we employed the PLS-DA model to determine VIP values. The reliability of this model was confirmed via a 200-permutation test (autumn: R2 = 0.0, 0.274, Q2 = −0.199; Figure 5F). Figure 5G shows that all four N application treatments are within the confidence interval and have good repeated clustering. The autumn PLS-DA model identified ten discriminating aroma compounds (VIP > 1; Figure 5H), encompassing four alcohols, two ketones, two alkenes, one hydrocarbon, and one ester (Table 1). These results echoed the findings of the PCA model, emphasizing similarities among N application treatments and their significant difference with the N0 group. According to Table 1, three aromatic compounds with OAVs exceeding 1 were identified in autumn samples. Linalool emerged as the most influential volatile with an OAV surpassing 1000, accompanied by three additional species with OAV > 1. The aroma profile of autumn JX primarily comprised linalool, (E)-2-hexenal, and 2,6,6-trimethyl-1-cyclohexene-1-carboxaldehyde, with the N0 treatment exhibiting the highest number of aroma species and concentrations (Table 2). These volatile compounds significantly contributed to the fundamental aroma characteristics of fresh JX leaves during autumn, with linalool serving as the primary aroma contributor.

## 4. Conclusions

This study applied nitrogen at four different levels (N0, N150, N300, N450) to Jin Xuan tea varieties during summer and autumn. The results revealed the identification of 49 volatile components, with aldehydes and alcohols dominating. The volatile compounds predominantly exhibited grassy and floral aromas, complemented by fruity, sweet, and woody notes.

Across seasons, the total concentration of volatile substances decreased without nitrogen application (N0). Conversely, the total concentration of volatiles increased with all nitrogen application levels (N150, N300, N450). The disparity between no-nitrogen and nitrogen application treatments diminished with a rise in the concentration of volatile substances overall, albeit with a decrease in the concentration of volatiles creating the floral aroma. Considering the main volatile types and key substances of black and oolong teas, it was found that fresh Jin Xuan leaves in summer and autumn were apt for producing both types of tea, with summer tea being more favorable for black tea. Comparing different application treatments, nitrogen application decreased the overall concentration of volatile substances in both seasons. The N0 treatment exhibited the highest concentration of volatile substances, the most robust overall aroma, and the widest range of substances. The overall concentration decreased from N0 to N300 in summer, then rose again at N450. However, there was no significant difference in overall concentration among N150, N300, and N450 in autumn. Regarding suitability for tea production, the N0 treatment in summer was deemed more suitable for producing black tea and oolong tea compared to the N0 and N450 treatments in autumn. Based on odor activity values and multivariate statistical analysis, (E)-2-hexenal, linalool, and geraniol emerged as the primary contributors to the aroma of fresh Jin Xuan leaves. These findings suggest that urea application in summer and autumn can decrease the content of volatile substances in Jin Xuan, with the most significant effect observed in summer. This study offers a scientific foundation for understanding the impact of seasonal nitrogen fertilizer application on tea aroma quality. Exploring gene expression in fresh Jin Xuan leaves under different seasons and nitrogen treatments is a future research direction.

## Figures and Tables

**Figure 1 foods-13-01776-f001:**
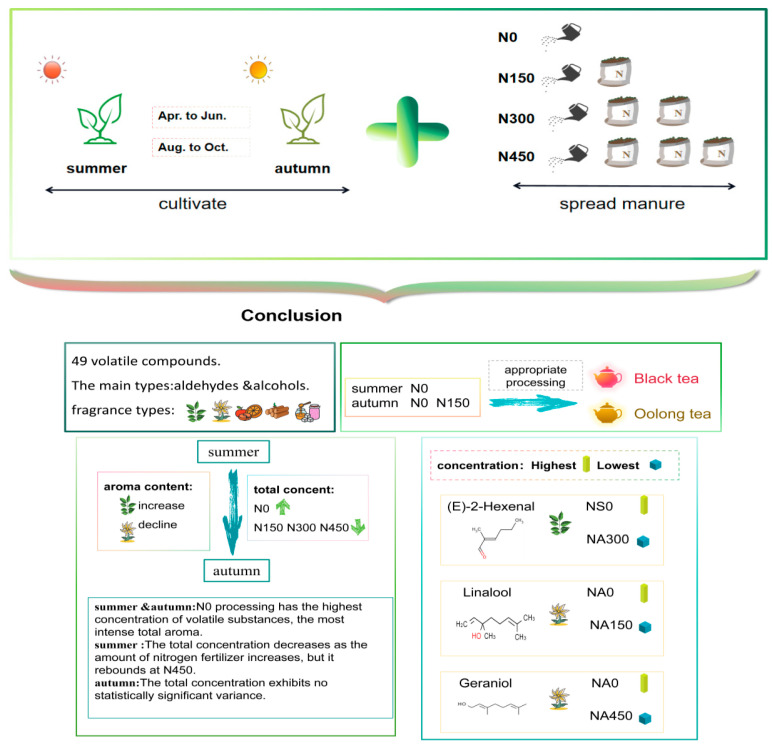
This study illustrates the effects of planting at four N application levels in summer and autumn on the volatiles of harvested fresh Jin Xuan leaves. NS0, NS150, NS300, and NS450 are N application treatments in summer, while NA0, NA150, NA300, and NA450 are N application treatments in autumn. NS0, NS150, NS300, and NS450 are N application treatments in summer, while NA0, NA150, NA300, and NA450 are N application treatments in autumn.

**Figure 2 foods-13-01776-f002:**
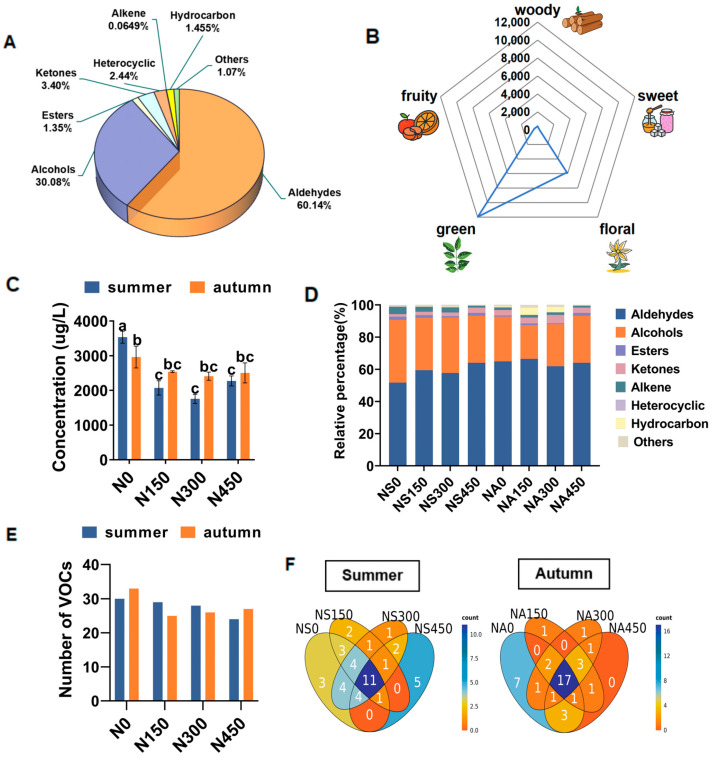
(**A**) The proportion of total volatile components in all treatments of fresh Jin Xuan (JX) leaves. (**B**) Radar image of total volatile aroma compounds in all treatments of fresh JX leaves. (**C**) The total concentration of volatile compounds in fresh JX leaves under four N application levels in summer and autumn. (**D**) The proportion of volatile species in fresh JX leaves under four N application levels in summer and autumn. (**E**) The number of volatile organic compounds (VOCs) in fresh JX leaves under four N application levels in summer and autumn. (**F**) Venn diagram of the four N application levels and volatile species in fresh JX leaves under different conditions in summer and autumn. Different letters indicate significant differences, where *p* ≤ 0.05.

**Figure 3 foods-13-01776-f003:**
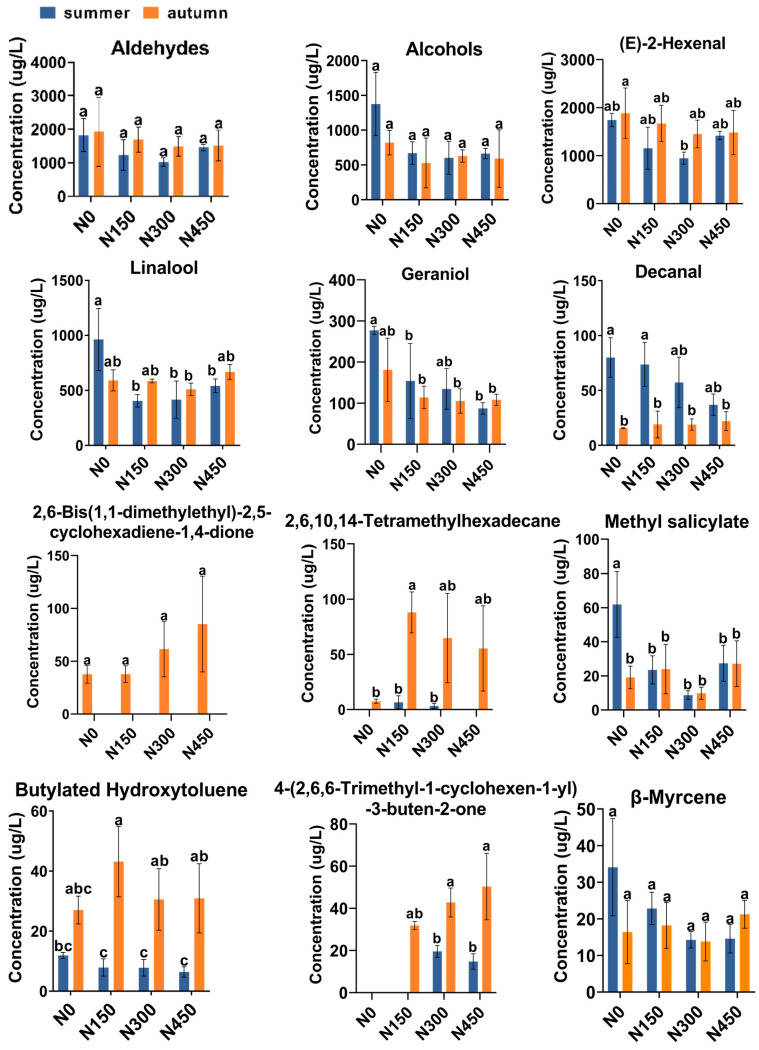
Dynamic changes of the 10 volatile compounds with the highest concentrations in fresh Jin Xuan leaves under four N application treatments (N0, N150, N300, N450) in summer and autumn. Different letters indicate significant differences, where *p* ≤ 0.05.

**Figure 4 foods-13-01776-f004:**
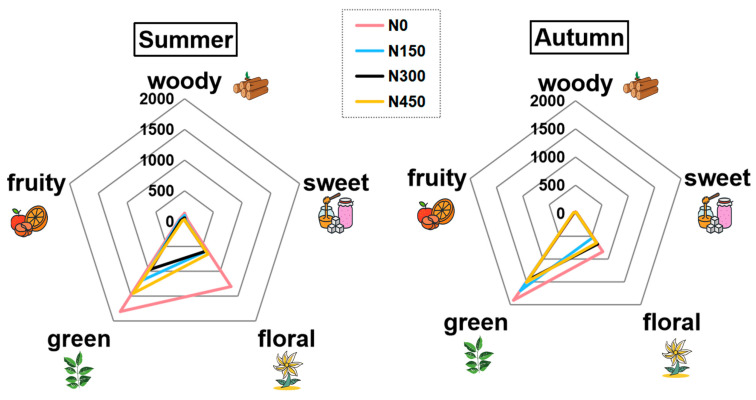
Aroma radar maps of fresh Jin Xuan (JX) leaves in summer and autumn under different N application levels (N0, N150, N300, N450), categorized into the five main aroma type dimensions—woody, sweet, floral, green, and fruity.

**Figure 5 foods-13-01776-f005:**
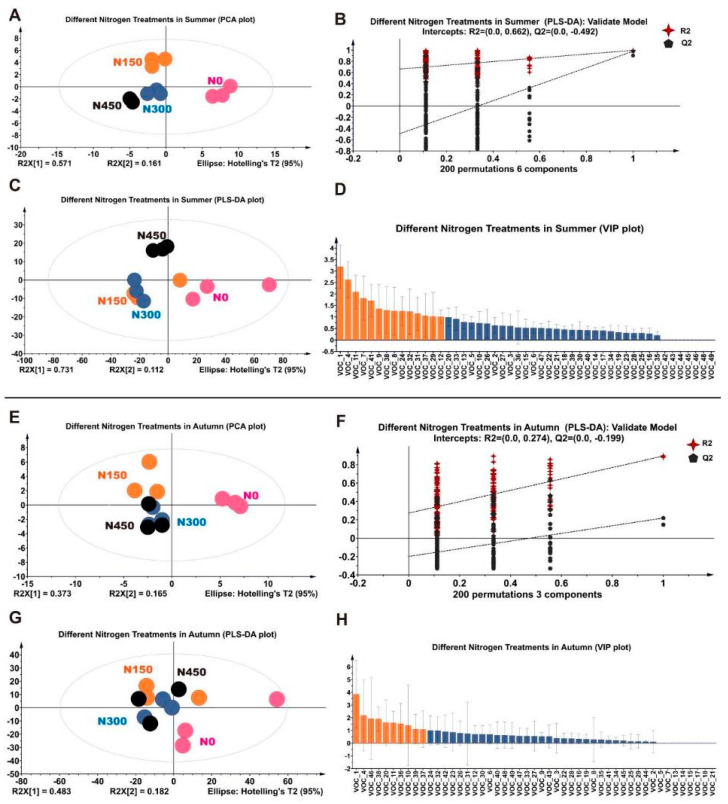
Multivariate statistical analysis of volatile data. (**A**,**E**) Summer (R2 = 0.9230, Q2 = 0.769) and autumn (R2 = 0.6560, Q2 = 0.076) principal component analysis (PCA) score plot. (**B**,**F**) Cross–validation results of 200 permutations (summer: R2 = 0.0, 0.662, Q2 = −0.492; autumn: R2 = 0.0, 0.274, Q2 = −0.199). (**C**,**G**) Partial least squares discriminant analysis (PLS–DA) for summer (R2X = 0, Q2 = 0.992) and autumn (R2X = 0, Q2 = 0.656). (**D**,**H**) Variable importance in projection (VIP) rating charts, where the orange bars indicate volatile compounds with VIP scores > 1 and the blue bars indicate volatiles with VIP scores < 1.

**Table 2 foods-13-01776-t002:** Odor activity analysis (OAV) analysis of volatile compounds in fresh Jin Xuan leaves at four N application levels in autumn.

Var. No.	Volatile Compounds	Odor Type	VIP ^1^	OT ^2^ (µg/kg)	OAVs in N Treatments			
					N0	N150	N300	N450
Var_1	(E)-2-hexenal	Fresh, fruity, green	3.30	110 ^a^	17.14	98.23	85.32	87.13
Var_4	Linalool	Floral, sweet	2.41	0.22 ^b^	2686.59	1778.55	2322.23	2098.82
Var_37	2,6,6-trimethyl-1-cyclohexene-1-carboxaldehyde	Herbal, clean, rose-like, fruity	1.27	5 ^d^	3.46	0.00	3.10	1.88
Var_3	β-Ocimene	Sweet, floral	1.16	10 ^h^	0.00	0.00	0.74	0.00
Var_10	(Z)-2,6-Octadien-1-ol, 3,7-dimethyl-	Fresh, citrus, floral, green, lemon-like	1.14	53 ^d^	0.18	0.74	0.13	0.17
Var_8	Methyl salicylate	Peppermint, minty, fresh, sweet	1.06	40 ^a^	0.48	0.60	0.25	0.68

^1^ VIP, variable importance in projection; ^2^ OT, odor threshold in water reported elsewhere, indicated as a,b,d,h [36,37,39,43].

## Data Availability

The original contributions presented in the study are included in the article/Appendix A, further inquiries can be directed to the corresponding author.

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
