# Peer review of "The Effects of Nitrogen Fertilizer on the Aroma of Fresh Tea Leaves from Camellia sinensis cv. Jin Xuan in Summer and Autumn"

_foods, 2024, doi:10.3390/foods13111776_

Round 1

Reviewer 1 Report

Comments and Suggestions for Authors

Dear Authors,

your work deal with Nitrogen supplementation on tea plants and qualitative and quantitative composition of their volatile compounds. Due the importance of the VOCs on the commercial value of this plant, your work is of good scientific interest. Results support the hypothesis and all experimental protocols are adequately described.

Howevar at least two parts of the manuscript should be deeply investigated and/or improved. 

The first is that in the discussion an indication on the effects of nitrogen integration on the quantity of biomass produced should be reported. The higher organoleptic quality of some samples should be accompanied by a comment on the biomass produced under the experimental conditions for a further validation, also economic, of the results.

The second is that in the table of VOCs composition, some deviation standards reported are really too high, insinuating doubt on the robustness of the data. Authors should clarify if these number can be refined or are results of a little number of replicates which, among other things, are not reported in the material and methods section and should be added.

Author Response

Thank you for your time and effort in reviewing my manuscript ,and for providing me with feedback . I appreciate the opportunity to revise my manuscript and for the constructive criticism.

Point-by-point responses to comments.

Your work deal with Nitrogen supplementation on tea plants and qualitative and quantitative composition of their volatile compounds. Due the importance of the VOCs on the commercial value of this plant, your work is of good scientific interest. Results support the hypothesis and all experimental protocols are adequately described.

Howevar at least two parts of the manuscript should be deeply investigated and/or improved. 

- The first is that in the discussion an indication on the effects of nitrogen integration on the quantity of biomass produced should be reported. The higher organoleptic quality of some samples should be accompanied by a comment on the biomass produced under the experimental conditions for a further validation, also economic, of the results.

R1: Thank you for your comment. We have now revised the manuscript and included the effects of nitrogen integration on the quantity of biomass produced (Lines 301-305) , as referred to in section 3.4. Thank you again for your careful review of our paper.

- The second is that in the table of VOCs composition, some deviation standards reported are really too high, insinuating doubt on the robustness of the data. Authors should clarify if these number can be refined or are results of a little number of replicates which, among other things, are not reported in the material and methods section and should be added.

R2: Thank you for your suggestion. We have now updated the manuscript and included the number of repetitions of experimental data (Lines 97), as referred to in section 2.3. Thank you for your valuable feedback and for helping us improve the quality of our manuscript.

Reviewer 2 Report

Comments and Suggestions for Authors

The manuscript entitled "The Effects of Nitrogen Fertilizer on the Aroma of Fresh Tea Leaves from Camellia sinensis cv. Jin Xuan in Summer and Autumn", from the authors Ansheng Li, Zihao Qiu, Jinmei Liao, Jiahao Chen, Wei Huang, Jiyuan Yao, Xinyuan Lin,Yuwang Huang,Binmei Sun, Shaoqun Liu and Peng Zheng.

The manuscript is interesting from the researcher's point of view, but some clarifications are needed.

Lines 166 and 167 overlap with Figure 1 so the text in lines 166 and 167 cannot be read.

In Figure 1, the marks NS0, NS150, NS300, NS450, NA0, NA150, NA300 and NA450 are used. The symbols "S" and "A" in the samples are explained later in the caption of Figure 5. It is customary to explain the symbols when they first appear in the text. I ask the authors to clarify these markings on the examined samples in in the caption of Figure 1.

Author Response

Thank you for your time and effort in reviewing my manuscript ,and for providing me with feedback . I appreciate the opportunity to revise my manuscript and for the constructive criticism.
Point-by-point responses to comments.

The manuscript entitled "The Effects of Nitrogen Fertilizer on the Aroma of Fresh Tea Leaves from Camellia sinensis cv. Jin Xuan in Summer and Autumn", from the authors Ansheng Li, Zihao Qiu, Jinmei Liao, Jiahao Chen, Wei Huang, Jiyuan Yao, Xinyuan Lin,Yuwang Huang,Binmei Sun, Shaoqun Liu and Peng Zheng.

The manuscript is interesting from the researcher's point of view, but some clarifications are needed.

- Lines 166 and 167 overlap with Figure 1 so the text in lines 166 and 167 cannot be read.

R1: Thank you for your comment. We have now revised the manuscript (Lines 179-180), as referred to in section 3.1. Thank you again for your careful review of our paper.

- In Figure 1, the marks NS0, NS150, NS300, NS450, NA0, NA150, NA300 and NA450 are used.

The symbols "S" and "A" in the samples are explained later in the caption of Figure 5. It is customary to explain the symbols when they first appear in the text. I ask the authors to clarify these markings on the examined samples in in the caption of Figure 1.

R2: Thank you for your suggestion. We have now updated the manuscript (Lines 11-14), as referred to in Figure 1. Thank you for your valuable feedback and for helping us improve the quality of our manuscript.

Reviewer 3 Report

Comments and Suggestions for Authors

The manuscript presents a comprehensive analysis and a significant contribution to the field of sensory properties and quality of tea, providing valuable information for understanding the complex interactions between plant chemistry and environmental factors.

My comments are as follows.

Brief data on the research methodology used and the conclusion sentence should be added to the abstract.

More data on the Jin Xuan variety should be added in the introduction. Is there any advantage of this variety compared to others?

Line 5-6:  “Young buds and leaves…..making it one of the most popular non-alcoholic caffeinated beverages”. Rephrase it.

The last paragraph in the Introduction should provide a hypothesis and highlight the novelty of this research.

Line 50: add city and country for the manufacturer of standards

Subtitles 3.3. and 3.4. “Effects of Different N Application Levels”  rephrase it.

Considering presenting Figure 5 as a graphical abstract

Add further direction(s) at the end of the manuscript.

Author Response

Thank you for your time and effort in reviewing my manuscript ,and for providing me with feedback . I appreciate the opportunity to revise my manuscript and for the constructive criticism.

Point-by-point responses to comments.

The manuscript presents a comprehensive analysis and a significant contribution to the field of sensory properties and quality of tea, providing valuable information for understanding the complex interactions between plant chemistry and environmental factors.

My comments are as follows.

-Brief data on the research methodology used and the conclusion sentence should be added to the abstract.

R1: Thank you for your comment. We have now revised the manuscript and included the addition of experimental methods and conclusions (abstract). Thank you again for your careful review of our paper.

-More data on the Jin Xuan variety should be added in the introduction. Is there any advantage of this variety compared to others?

R2: Thank you for your comment. We have now revised the manuscript and included the variety advantages of Jinxuan (Lines 19-20), as referred to in 1. Introduction. Thank you again for your careful review of our paper.

-Line 5-6:  “Young buds and leaves…..making it one of the most popular non-alcoholic caffeinated beverages”. Rephrase it.

R3: Thank you for your comment. We have now revised the manuscript and included a restatement of this sentence (Lines 17-19), as referred to in 1. Introduction. Thank you again for your careful review of our paper.

-The last paragraph in the Introduction should provide a hypothesis and highlight the novelty of this research.

R4: Thank you for your comment. We have now revised the manuscript and included the addition of assumptions (Lines 53-59), as referred to in 1. Introduction. Thank you again for your careful review of our paper.

-Line 50: add city and country for the manufacturer of standards

R5: Thank you for your comment. We have now revised the manuscript (Lines 63), as referred to in 2.1. Thank you again for your careful review of our paper.

-Subtitles 3.3. and 3.4. “Effects of Different N Application Levels”  rephrase it.

R6: Thank you for your comment. We have now revised the manuscript and included a restatement of this sentence (Lines 63), as referred to in 2.1. Thank you again for your careful review of our paper.

-Considering presenting Figure 5 as a graphical abstract

R7: Thank you for your comment. We have now revised the manuscript to include Figure 5 as a graphical abstract (Figure 1). Thank you again for your careful review of our paper.

-Add further direction(s) at the end of the manuscript.

R8: Thank you for your comment. We have now revised the manuscript and included the addition of outlook directions in the conclusion (412-413), as referred to in 4. Conclusions. Thank you again for your careful review of our paper.

Round 2

Reviewer 1 Report

Comments and Suggestions for Authors

Dear Authors,

thank you for taking int account my suggestions to improve your manuscript.